# Effects of *Ulva* sp. Extracts on the Growth, Biofilm Production, and Virulence of Skin Bacteria Microbiota: *Staphylococcus aureus*, *Staphylococcus epidermidis*, and *Cutibacterium acnes* Strains

**DOI:** 10.3390/molecules26164763

**Published:** 2021-08-06

**Authors:** Mathilde Fournière, Gilles Bedoux, Djouhar Souak, Nathalie Bourgougnon, Marc G. J. Feuilloley, Thomas Latire

**Affiliations:** 1Laboratoire de Biotechnologie et Chimie Marines LBCM EA 3884, IUEM, Université Bretagne Sud, 56000 Vannes, France; gilles.bedoux@univ-ubs.fr (G.B.); nathalie.bourgougnon@univ-ubs.fr (N.B.); tlatire@uco.fr (T.L.); 2Université Catholique de l’Ouest Bretagne Nord, 22200 Guingamp, France; 3Laboratoire de Microbiologie Signaux et Microenvironnement LMSM EA4312, Université de Rouen Normandie, 27000 Évreux, France; djouhar.souak@univ-rouen.fr (D.S.); marc.feuilloley@univ-rouen.fr (M.G.J.F.)

**Keywords:** seaweed, ulvan, EAE, *Staphylococcus aureus*, *Staphylococcus epidermidis*, *Cutibacterium acnes*, biofilm, inflammation

## Abstract

*Ulva* sp. is known to be a source of bioactive compounds such as ulvans, but to date, their biological activity on skin commensal and/or opportunistic pathogen bacteria has not been reported. In this study, the effects of poly- and oligosaccharide fractions produced by enzyme-assisted extraction and depolymerization were investigated, for the first time in vitro, on cutaneous bacteria: *Staphylococcus aureus*, *Staphylococcus epidermidis*, and *Cutibacterium acnes*. At 1000 μg/mL, poly- and oligosaccharide fractions did not affect the growth of the bacteria regarding their generation time. Polysaccharide *Ulva* sp. fractions at 1000 μg/mL did not alter the bacterial biofilm formation, while oligosaccharide fractions modified *S. epidermidis* and *C. acnes* biofilm structures. None of the fractions at 1000 μg/mL significantly modified the cytotoxic potential of *S. epidermidis* and *S. aureus* towards keratinocytes. However, poly- and oligosaccharide fractions at 1000 μg/mL induced a decrease in the inflammatory potential of both acneic and non-acneic *C. acnes* strains on keratinocytes of up to 39.8%; the strongest and most significant effect occurred when the bacteria were grown in the presence of polysaccharide fractions. Our research shows that poly- and oligosaccharide *Ulva* sp. fractions present notable biological activities on cutaneous bacteria, especially towards *C. acnes* acneic and non-acneic strains, which supports their potential use for dermo-cosmetic applications.

## 1. Introduction

The seaweed *Ulva* sp. (Chlorophyta, Ulvales, *Ulvaceae*) is the source of large-scale green tides every year on the Brittany coastline (France) due to its proliferation as a result of marine eutrophication [1,2,3,4]. These green algal blooms have a profound detrimental ecological impact, causing an alteration of the ecosystem structure and a decrease in indigenous biodiversity [5,6], but they also have a considerable economic impact [1]. The valorization of *Ulva* sp. for human healthcare is currently minimal. This seaweed from stranding events is principally used as a soil amendment or animal feed; it is also not valorized and simply degraded by decay or combustion [1]. *Ulva* sp. is an important feedstock for structural polysaccharides known as ulvans, which can represent up to 36% of its dry weight [7]. Ulvans are sulfated cell-wall matrix polysaccharides, mainly composed of rhamnose, uronic acids (glucuronic and iduronic), and xylose. The constitutive repeating disaccharide units of the ulvans are mostly aldobiuronic acids (ulvanobiuronic acids) and minority aldobioses (ulvanobioses). Aldobiuronic acids are divided into type A_3_S (β-d-glucuronic acid (1,4)-linked to α-l-rhamnose 3-sulfate) and type B_3_S (α-l-iduronic acid (1,4)-linked to α-l-rhamnose 3-sulfate). Ulvanobioses consist of sulfated or non-sulfated β-d-xylose (1,4)-linked to α-l-rhamnose 3-sulfate [8]. Ulvans are traditionally extracted by hot-water extraction at high temperatures (80–90 °C), known as a maceration procedure, followed by an ethanolic precipitation [7]. Novel technology called enzyme-assisted extraction (EAE) can be employed to extract ulvans, as it shows various advantages such as a high extraction yield, mild reaction conditions, the preservation of biological activities, and reduced energy consumption [9,10,11,12,13].

The skin is the largest organ of the human body; it is made up of overlapping layers organized in three major components: epidermis, dermis, and hypodermis. Keratinocytes are the main cells of the epidermis, and fibroblasts are the main cells of the dermis [14]. Ulvans have demonstrated in vitro biological activities towards skin components such as human skin fibroblasts [13,15,16,17,18], which can be related to the structural features of the skin (degree of sulfation, sulfation pattern, monosaccharide composition, molecular weight, and glycosidic linkage) [15,16].

Over the last 10 years, the skin microbiota has been integrated as an essential constitutive element of the cutaneous system. This microbiota is defined as the microorganisms living in or on the skin ecosystem [19] (e.g., 1 million/cm^2^), and it comprises bacteria, fungi [20,21], yeasts, viruses [22], archaea [23], and mites [24]. It mainly contributes to the maintenance of a healthy skin, and its dysbiosis is a source of inflammatory skin diseases [25,26,27,28]. Within the skin microbiota, it appears that some bacterial species such as Gram-positive bacteria *Staphylococcus epidermidis*, *Staphylococcus aureus* (Firmicutes phylum), and *Cutibacterium acnes* (formerly *Propionibacterium acnes* [29], Actinobacteria phylum) prevail at skin level, especially in moist and sebaceous areas [25,30,31,32]. *S. aureus* and *S. epidermidis* are aerobic bacteria mostly found in external layers of the stratum corneum [33], whereas anaerobic *C. acnes* is mostly found in more internal layers or skin appendages such as hair follicles and sebaceous glands [34]. *C. acnes* is highly heterogeneous, with six main phylotypes (IA1, IA2, IB, IC, II, and III) [35,36] and 10 principal ribotypes (RT1 to RT10) [37]. Fitz-Gibbon et al. (2013) showed that some *C. acnes* ribotypes are associated with acne, such as the RT4 strain, whereas the RT6 strain colonizes non-acneic skin [37]. Within its ecological niche, *C. acnes* is associated with *S. aureus*, and both contribute to a healthy microbiota [38]. *S. aureus*, *S. epidermidis*, and *C. acnes* play a fundamental role in the skin microbiota due to their commensal character and their contribution to skin homeostasis. However, these bacteria can turn to opportunistic pathogens depending upon the skin microenvironment [30,39,40,41]. In case of pathogenicity, these bacterial strains can be found in a biofilm form that offers strong resistance to antimicrobial compounds and to the action of an immune skin system [42,43,44].

Both research progress and European Union regulations have motivated the examination of the potential impact of dermo-cosmetic active ingredients on the cutaneous microbiota. The maintenance, protection, and restoration of the skin microbiota’s diversity and equilibrium, and the prevention of skin dysbiosis, constitute one of the research tracks in the cosmetic industry [30,45,46,47]. Active ingredients can modulate bacterial microflora by inducing or repressing bacterial metabolic pathways and can thus influence the skin homeostasis [30,48]. The main activities of ingredients targeted towards the skin bacteria are (1) the promotion or respect of commensal metabolism and/or bacterial diversity in order to limit invasion by pathogens, (2) the reduction of pathogen bacteria growth, virulence, and biofilm formation, and (3) the modulation of the skin microenvironment and immune system responses in case of skin dysbiosis [30].

Marine-derived poly- and oligosaccharides are often considered as potential prebiotics, corresponding to nutrients beneficial to health, for skin commensal bacteria [49]. Thus, seaweeds rich in poly-/oligosaccharides are viewed as potential prebiotics for the skin microbiota [50]. Recent studies have shown that cosmetic ingredients composed of seaweed extracts were able to restore a healthy microbiome and reinforce microbial diversity following stress-induced dysbiosis [51], or increase bacterial diversity in reactive and sensitive skins [52]. To date, however, no study has been conducted on the effects of ulvans or their derivatives on the skin microbiota.

Hence, this study investigates for the first time, in vitro, the effects of poly- and oligosaccharide fractions from *Ulva* sp. obtained using the novel green technology EAE on four cutaneous bacteria: *Staphylococcus aureus* MFP03 and *Staphylococcus epidermidis* MFP04, collected on normal human skin [53], and *Cutibacterium acnes* RT4 (HL045PA1/HM-516, “acneic” strain) and RT6 (HL110PA3/HM-554, “non-acneic” strain), initially isolated by Fitz-Gibbon et al. (2013) [37]. In this study, the effects of *Ulva* poly- and oligosaccharides on their growth, biofilm production, cytotoxicity, and/or inflammatory potential on keratinocytes are presented.

## 2. Results

### 2.1. Poly- and Oligosaccharide Ulva sp. Fractions: Biochemical Composition and Molecular Weight Distribution

The detailed biochemical composition analysis (carbohydrates, uronic acids, sulfate groups, and proteins), monosaccharide determination (rhamnose, glucuronic acid, and xylose), and molecular weight distribution of *Ulva* sp. fractions derived from EAE have already been reported [13,16].

The relevant data for the present study are then summarized. Polysaccharide fractions of crude ulvans (UE) and dialyzed ulvans (DS-UE) are composed of high-molecular-weight ulvans (Mw > 670 kDa). Their composition is characterized by 23.5–37.4% carbohydrates (rhamnose 50.2–50.4%, glucuronic acid 10.6–11.9%, and xylose 2.8–3.2%), 18.5–37.0% uronic acids, and 29.9–49.1% sulfate groups. The dialyzed fraction (DS-UE) is significantly richer in ulvans than the crude fraction (UE). Oligosaccharide fractions, depolymerized ulvans from radical hydrolysis by H_2_O_2_ (DEP-HD PP-UE), and depolymerized ulvans from acid hydrolysis by Amberlite resin (DEP-AD PP-UE) are constituted by low-molecular-weight ulvans (8 kDa for DEP-HD PP-UE and 1.5 kDa for DEP-AD PP-UE) with 24.4–30.4% carbohydrates (rhamnose 44.9–55.4% and glucuronic acid 7.5–11.0%) and 21.6–30.8% uronic acids. It is noteworthy that the depolymerization process conducted led to a partial loss of sulfates (not detected in DEP-AD PP-UE by Azure A method but identified in MALDI-TOF and only 6.6% in DEP-HD PP-UE).

### 2.2. Effects of Poly- and Oligosaccharide Ulva sp. Fractions on the Bacterial Growth Kinetic 

#### 2.2.1. Effects on the Growth of *S. aureus* MFP03 and *S. epidermidis* MFP04

The effects of poly- and oligosaccharide *Ulva* sp. fractions at 1000 μg/mL on the growth of *S. aureus* MFP03 (Figure 1a) and *S. epidermidis* MFP04 (Figure 1b) in a tryptic soy broth (TSB) culture medium were evaluated by monitoring over 24 h.

Poly- and oligosaccharide fractions had no effect on the growth of *S. aureus* MFP03 (Figure 1a) and *S. epidermidis* MFP04 (Figure 1b).

In addition, the generation times of the bacteria (Figure 2) in a TSB culture medium, which were 27.8 ± 0.6 and 40 ± 0.3 min for *S. aureus* MFP03 and *S. epidermidis* MFP04, respectively, were statistically unchanged when the bacteria were exposed to poly- and oligosaccharide *Ulva* sp. fractions in comparison with the control.

#### 2.2.2. Effects on the Growth of Acneic and Non-Acneic *C. acnes* Strains

The effects of poly- and oligosaccharide *Ulva* sp. fractions at 1000 μg/mL on the growth of the acneic *C. acnes* RT4 strain (Figure 3a) in a reinforced clostridial medium (RCM) pastone +2% glucose and non-acneic *C. acnes* RT6 strain (Figure 3b) in RCM polypeptone +2% glucose were evaluated by monitoring over 72 h.

The growth of *C. acnes* RT4 was unchanged when exposed to polysaccharide fractions (UE and DS-UE) and oligosaccharide fraction DEP-AD PP-UE regarding both the exponential phase (generation time) and the stationary phase (Figure 3a). However, the oligosaccharide fraction DEP-HD PP-UE induced a decrease in the stationary phase of the RT4 strain, causing a 50% reduction in the final biomass (Figure 3a). The growth of *C. acnes* RT6 did not show a noticeable difference in the presence of poly- and oligosaccharide fractions when considering both exponential and stationary phases (Figure 3b). However, important variations were noted during the growth phase, probably because of the limited number of replicates and the harsh growth conditions of this strain, which is strongly anaerobic and hydrophobic [54].

In addition, the generation times of the bacteria (Figure 4), which were 4.4 ± 0.1 h for *C. acnes* RT4 and 6.1 ± 1.5 h for *C. acnes* RT6, were unchanged when the bacteria were exposed to poly- and oligosaccharide *Ulva* sp. fractions.

### 2.3. Effects of Poly- and Oligosaccharide Ulva sp. Fractions on Biofilm Production

Biofilms visualized by confocal laser scanning microscopy were submitted to image analysis in order to determine the biovolume (μm^3^/μm^2^), which provides an estimation of the biomass and mean thickness (μm). Results are expressed as a percentage of variation in relation to the control (i.e., bacteria grown in the absence of *Ulva* extracts).

#### 2.3.1. Effects on the Biofilm Production of *S. aureus* MFP03 and *S. epidermidis* MFP04

*S. aureus* MFP03 (Figure 5a) and *S. epidermidis* MFP04 (Figure 5b) preliminarily cultured in TSB in the absence (control) or presence of poly- and oligosaccharide *Ulva* sp. fractions (1000 μg/mL) were allowed to form mature biofilms over 48 h in static aerobic conditions.

*S. aureus* MFP03 biofilms formed by control and treated bacteria had identical mean thickness (4.5 ± 0.5 μm) and biovolume (3.4 ± 0.5 μm^3^/μm^2^) values (Figure 5a). In addition, biofilms showed similar 3D organization and density (Figure 5c). No significant modification of the biofilm structure of *S. aureus* MFP03 was induced by poly- and oligosaccharide *Ulva* sp. fractions (Figure 5a).

*S. epidermidis* MFP04 biofilms generated in the same conditions were observed by confocal laser scanning microscopy. In the control conditions, the mean thickness of the biofilms reached 7.1 ± 0.6 μm and the biovolume 4.6 ± 0.4 μm^3^/μm^2^ (Figure 5d). No significant modification of the biofilms formed by *S. epidermidis* MFP04 was induced by the polysaccharide fractions (UE and DS-UE) and oligosaccharide fraction DEP-AD PP-UE (Figure 5b). However, a significant increase in the biovolume (+10.6%; *p* < 0.05) and mean thickness (+14.7%; *p* < 0.01) was noted when the bacteria were exposed to the oligosaccharide fraction DEP-HD PP-UE (Figure 5b,d).

#### 2.3.2. Effects on the Biofilm Production by Acneic and Non-Acneic *C. acnes* Strains

Acneic *C. acnes* RT4 (Figure 6a) and non-acneic *C. acnes* RT6 (Figure 6b) preliminarily cultured in the absence (control) or presence of poly- and oligosaccharide *Ulva* sp. fractions (1000 μg/mL) were allowed to form mature biofilms over 72 h in static anaerobic conditions.

*C. acnes* RT4 biofilms formed in control conditions reached a mean thickness of 9.5 ± 1.8 μm and a biovolume of 5.9 ± 1.8 μm^3^/μm^2^ (Figure 6c). No significant modification of the biofilm structure was induced by poly- and oligosaccharide *Ulva* sp. fractions, although decreases in biovolume (−16.0%) and mean thickness (−20.9%) were measured after exposure to DEP-AD PP-UE (Figure 6a).

Control *C. acnes* RT6 (Figure 6d) biofilms observed in the same conditions had a mean thickness of 5.2 ± 1.3 μm and a biovolume of 3.4 ± 0.9 μm^3^/μm^2^. No significant modification of the biofilm structure was induced by the polysaccharide (UE and DS-UE) and oligosaccharide fractions DEP-HD PP-UE (Figure 6b). However, a marked decrease (*p* < 0.01) in biovolume (−45.7%) and mean thickness (−44.1%) was induced by exposure of the bacteria to the oligosaccharide fraction DEP-AD PP-UE (Figure 6b,d).

### 2.4. Effects of Poly- and Oligosaccharide Ulva sp. Fractions on the Cytotoxic Activity of S. aureus MFP03 and S. epidermidis MFP04 towards HaCaT Keratinocytes

The bacteria *S. aureus* MFP03 and *S. epidermidis* MFP04 were pretreated with poly- and oligosaccharide *Ulva* sp. fractions at 1000 μg/mL before infecting HaCaT keratinocytes. The effects of the fractions on the cytotoxic activity of *S. aureus* MFP03 and *S. epidermidis* MFP04 towards HaCaT keratinocytes were assessed by measuring the LDH (lactate dehydrogenase) release using a Pierce^TM^ LDH Cytotoxicity kit assay (Figure 7). The results were expressed as a percentage of variation in comparison to the control (bacteria grown in TSB alone without exposure to *Ulva* extracts).

The cytotoxicity assay (Figure 7) showed that none of the poly- or oligosaccharide fractions significantly modified the cytotoxic potential of either strains (*S. aureus* MFP03 or *S. epidermidis* MFP04) towards HaCaT keratinocytes.

### 2.5. Effects of Poly- and Oligosaccharide Ulva sp. Fractions on the Inflammatory Potential of Acneic and Non-Acneic C. acnes Strains towards HaCaT Keratinocytes

The intrinsic cytotoxic activity of *C. acnes* RT4 and RT6 on keratinocytes is negligible [54]. Hence, the effects of *Ulva* extracts centered on the inflammatory activity of these bacteria. The induction of inflammation was evaluated by an assay of interleukin-8 (IL-8) released by HaCaT keratinocytes after exposure to the bacteria (Figure 8). Results were normalized as a percentage of variation of IL-8 concentration measured in the cell culture medium after incubation in the presence of control (untreated) RT4 or RT6 *C. acnes*.

The basal release of IL-8 by HaCaT keratinocytes was low (1.8 ± 0.3 pg/mL). Both *C. acnes* strains had a stimulatory effect on the IL-8 release by keratinocytes, and the acneic strain RT4 showed higher inflammatory potential; the IL-8 concentration in the culture medium reached 1051.2 ± 283.5 pg/mL at the end of the experiment and 801.6 ± 203.3 pg/mL after interaction with the non-acneic strain RT6 (data not shown).

The release of IL-8 by keratinocytes induced by the acneic *C. acnes* RT4 strain was significantly decreased (*p* < 0.05) when the strain was cultured in the presence of polysaccharide *Ulva* sp. fractions (−39.9% for UE and −36.1% for DS-UE) (Figure 8). The oligosaccharide *Ulva* sp. fractions also induced a decrease in IL-8 release by keratinocytes after infection with RT4, but the difference from the control was not statistically significant (−25.9% for DEP-HD PP-UE and −12.1% for DEP-AD PP-UE). The release of IL-8 by keratinocytes induced by the non-acneic *C. acnes* RT6 strain was significantly decreased after the strain was cultured in the presence of polysaccharide *Ulva* sp. fractions (−57.0% for UE and −58.3% for DS-UE; *p* < 0.01). In addition, the oligosaccharide *Ulva* sp. fraction DEP-HD PP-UE also induced a significant decrease (−18.3%; *p* < 0.05) in IL-8 release by keratinocytes after infection with RT6, while DEP-AD PP-UE induced a minor non-significant decrease.

## 3. Discussion

Our main goal was to evaluate for the first time the potential biological activities on model cutaneous bacteria (i.e., *S. epidermidis*, *S. aureus*, and *C. acnes*) of poly- and oligosaccharide fractions from *Ulva* sp. obtained after EAE, which is known to improve extraction yield and ulvan enrichment in comparison with maceration [13,55].

The effects of *Ulva* fractions on the bacteria were investigated by measuring growth, biofilm production, and cytotoxic or inflammatory potentials towards HaCaT keratinocytes. Because of the high content of poly- and oligosaccharide *Ulva* fractions in carbohydrates (50% rhamnose) and uronic acids (10% glucuronic acid), we postulate that the biological activities detected are mainly related to ulvan composition [7,16].

### 3.1. Effects of Poly- and Oligosaccharide Ulva sp.-Derived EAE Fractions on Commensal Staphylococcus Species and Acneic or Non-Acneic Cutibacterium acnes

Poly- and oligosaccharide *Ulva* sp. fractions at 1000 μg/mL had no effect on the growth of *S. aureus* MFP03 and *S. epidermidis* MFP04. Similarly, recent studies have demonstrated that polysaccharides from other seaweeds (*Gracilaria* sp., *Sargassum vachellianum*, and *Fucus spiralis*), eventually sulfated, and ulvans from *Ulva reticulata* do not exert an inhibitory effect on the growth of reference strains of *S. aureus* or *S. epidermidis* [56,57,58]. Because of its richness in carbohydrates, the composition of *Ulva* sp. poly- and oligosaccharides is close to that of PS291^®^ (Téflose^®^, SOLABIA GROUP), a rhamnose-rich polysaccharide (60%) that has previously been examined on the same strains of *S. aureus* and *S. epidermidis* (MFP03 and MFP04) and also had no effect on their growth [59,60].

Thus, our results regarding the growth of *S. aureus* and *S. epidermidis* in the presence of poly- and oligosaccharide *Ulva* sp. fractions are congruent with the literature data on the effects of seaweed polysaccharides and polysaccharides rich in rhamnose. Although the skin bacterial microbiota is dominated by *Staphylococcus* species such as *S. aureus* and *S. epidermidis*, *Cutibacterium acnes* is also significantly present [32]. Our study demonstrates that acneic strain RT4 and non-acneic strain RT6 do not have their generation time modified in the presence of poly- and oligosaccharide *Ulva* sp. fractions at 1000 μg/mL. This is consistent with Gannesen et al. (2018) who showed that the rhamnose-rich polysaccharide PS291^®^ also had minor effects on the growth kinetic of *C. acnes* RT4 [59].

In normal skin, *S. epidermidis*, *S. aureus*, and *C. acnes* usually behave as commensal strains in symbiosis with the cutaneous system [30,31], but in response to variations of the skin microenvironment [40,61,62,63,64], these bacteria can shift to a pathogen phenotype showing variations of biofilm formation and virulence expression [37,65,66,67,68].

As assessed by confocal laser scanning microscopy, the biofilm formation by *S. epidermidis* MFP04 was increased when the bacteria were cultured in the presence of DEP-HD PP-UE, while other fractions did not show any effect. In addition, no significant effect was observed on biofilm formation by *S. aureus* MFP03 cultured in the presence of poly- and oligosaccharide *Ulva* sp. fractions. To our knowledge, this is the first study investigating the effects of seaweed polysaccharides on the biofilm formation of *S.*
*aureus* and *S. epidermidis* strains isolated from a healthy skin. Using the crystal violet staining technique, Hillion (2013) showed that the rhamnose-rich polysaccharide PS291^®^ was capable of decreasing biofilm formation by *S. epidermidis* MFP04. Using confocal microscopy, Gannesen et al. (2018) demonstrated that this ingredient also decreased biofilm formation activity by *S. aureus* MFP03 [59,60]. However, Hillion (2013) also reported that another polysaccharide of high-molecular-weight (M_w_ = 2000 kDa) rich in galacturonic acid, L-fucose, and glucose had no effect on *S. epidermidis* MFP04 biofilm formation [60]. Thus, the effects of polysaccharides on biofilm formation by *S. epidermidis* MFP04 could be related to their monosaccharide composition. Moreover, Hillion (2013) also demonstrated that a low-molecular-weight oligorhamnose, obtained by hydrolysis of Rhamnosoft^®^ (a cosmetic polysaccharide ingredient of 60 kDa, rich in rhamnose, galactose, and glucuronic acid) had a positive effect on *S. epidermidis* MFP04 biofilm formation [60]. Our results with DEP-HD PP-UE of low M_w_ (8 kDa) and monosaccharide composition similar to Rhamnosoft^®^ are thus congruent with previous results.

The data on *C. acnes* biofilm production are relatively limited, and to date, no study has investigated the effects of seaweed polysaccharides on the biofilm formation by acneic and non-acneic forms of *C. acnes*. Our study demonstrates for the first time that *Ulva* sp.-derived polysaccharide fractions (UE and DS-UE) and oligosaccharide fraction DEP-HD PP-UE have no effect on biofilm formation by acneic and non-acneic strains *C. acnes*. Conversely, the oligosaccharide fraction DEP-AD PP-UE induced a marked decrease in biofilm formation by the non-acneic strain *C. acnes* RT6 (*p* < 0.01). A lower production of biofilm was also observed in the acneic strain *C. acnes* RT4, but the difference was not significant in relation to the control.

In previous studies, Enault et al. (2014) worked on a non-ribotyped strain of *C. acnes*, and Gannesen et al. (2018) worked on the acneic *C. acnes* RT4 strain. The authors showed an inhibitory effect of PS291^®^ on biofilm formation that was explained by its overly strong anti-adhesive properties that could limit the early steps of biofilm formation [59,69]. As in our study, this ingredient led to a modification of the *C. acnes* RT4 biofilm architecture and to a reduction of the mean thickness [59]. Nevertheless, our results are only partially congruent with the literature, since one of the *Ulva* oligosaccharide fractions showed antibiofilm activity, which suggests a certain specificity in the structure-activity relationships of the polymer.

We also show that poly- and oligosaccharide *Ulva* sp. fractions did not induce any modification of the cytotoxic potential of *S. aureus* MFP03 and *S. epidermidis* MFP04 towards HaCaT keratinocytes. To our knowledge, no previous study has investigated the modulation of the cytotoxic potential of these staphylococci after culture in the presence of seaweed polysaccharides. However, our results are congruent with Hillion (2013) who showed that PS291^®^ did not modify the cytotoxic potential of *S. aureus* MFP03 and *S. epidermidis* MFP04 towards HaCaT cells [60]. Because of their very limited cytotoxicity [54], this parameter was not investigated on *C. acnes* strains, and it was preferred to test the effects of *Ulva* extracts on the inflammatory potential of these bacteria [70,71]. IL-8 was used as a marker of *C. acnes*-induced inflammation in HaCaT keratinocytes. We observed that, as previously described [54], HaCaT keratinocytes are able to respond to both *C. acnes* strains (RT4 and RT6) with a marked increase in IL-8 release, and the RT4 acneic strain has a greater effect. Interestingly, *Ulva* sp. poly- and oligosaccharide fractions showed a marked anti-inflammatory potential by reducing IL-8 release by keratinocytes after exposure to RT4 (−39.9% maximal value) and RT6 (−58.3% maximum value) *C. acnes*. This anti-inflammatory activity may result from the binding of ulvan rhamnose residues to bacterial lectins and the inhibition of the bacterial adhesion to keratinocytes [72].

### 3.2. Potential Implications of the Sulfate Composition and Molecular Weight of Ulvan Fractions and Strain Specificity in Skin Bacteria Modulation

Our study demonstrates that *Ulva* sp. poly- and oligosaccharides have multiple effects on *S. aureus*, *S. epidermidis*, and *C. acnes* that are mostly strain-specific. Polysaccharide fractions with high-molecular-weight sulfated ulvans (UE and DS-UE) did not induce biofilm formation by *S. epidermidis* MFP04 and *S. aureus* MFP03, whereas oligosaccharide fraction DEP-HD PP-UE, with lower molecular weight (8 kDa) and partially desulfated ulvans, promoted biofilm formation by *S. epidermidis* MFP04. Similarly, only the oligosaccharide fraction DEP-AD PP-UE, which is composed of partially desulfated ulvans of the lowest M_w_ (1.5 kDa), showed an antibiofilm activity against *C. acnes* RT4. These results could be coincidental, but they could also be related to the higher efficiency of the desulfated forms of polysaccharides. Gadenne et al. (2015) showed that the best antifouling surface against an *S. aureus* strain was functionalized with a desulfated ulvan [73]. In addition to considering the inflammatory potential of *C. acnes* RT4 and RT6 towards HaCaT keratinocytes, we demonstrate that ulvan depolymerization, which leads to a decrease in molecular weight and sulfation degree, is associated with the weakest anti-inflammatory activity for oligosaccharides in comparison to polysaccharides. Thus, various parameters, including molecular weight and the poly- and oligosaccharide sulfation degree, influence the biological activities of *Ulva* sp. extracts on skin bacteria. On the other hand, we cannot neglect the presence of proteins in the poly- and oligosaccharide *Ulva* sp. fractions (10.5–16.8%), which suggests a potential influence of the protein/peptide content, amino acid composition, and even partial protein hydrolysis due to depolymerization treatments for *Ulva* sp. fractions production in modulation of the biological activities, as previously observed in fibroblasts [74].

## 4. Materials and Methods

### 4.1. Seaweed Material

The green seaweed *Ulva* sp. (Chlorophyta, Ulvales, *Ulvaceae*) was collected from the intertidal beach Landrézac (47°30′17.9″ N 2°42′37.1″ O) in Sarzeau (Brittany, France) on 28 May 2018, during the afternoon, at low tide, before stranding. The seaweed material was washed with tap water, ground to 3 mm in diameter, frozen at −25 °C, freeze-dried (Alpha 1–4 LSC, Martin Christ Gefriertrocknungsanlagen GmbH, Osterode am Harz, Germany), and stored at room temperature in the dark until use.

### 4.2. Poly- and Oligosaccharide Ulva sp. Fractions Production and Characterization

The detailed procedures of poly- and oligosaccharide *Ulva* sp. fractions production from EAE were fully described in Fournière et al. (2019) (see Figure 9 and Table 1) [13].

To summarize, enzymatic hydrolysis was performed at 50 °C for 3 h by using endo-protease Protamex^®^ at 6% (w/dw) in distilled water on freeze-dried *Ulva* sp. material; next, the enzyme was inactivated at 90 °C for 15 min. The hydrolysate was treated by ethanolic precipitation (absolute ethanol, 1:5, *v*/*v*) at 4 °C for 24 h in order to produce a fraction rich in crude ulvans (UE).

A purification step was then performed by dialysis of the fraction rich in crude ulvans (UE, 10 mg/mL) at 4 °C for 7 days (cut-off 12–14 kDa, Spectra/Por^®^4 Dialysis Membrane, Spectrum Laboratories, Fisher Scientific, Illkirch, France) in order to produce the fraction of dialyzed ulvans (DS-UE).

Depolymerization procedures, radical hydrolysis by hydrogen peroxide (H_2_O_2_), and acid hydrolysis by ion-exchange resin were applied on an ethanolic precipitate solution (25 mg/mL) obtained from UE. For radical depolymerization, H_2_O_2_ (8%, *v*/*v*, Fisher Scientific, Illkirch, France) was mixed with the solution at 50 °C for 24 h. For acid depolymerization, ion-exchange resin Amberlite^®^ FPC23 H (10 mL equivalent, Sigma-Aldrich, Saint Quentin Fallavier, France) was mixed with the solution at 80 °C for 24 h according to the protocol of Adrien et al. (2017), and the mixture after filtration was neutralized with NaOH (0.1 and 1 M) [15]. Next, both solutions from radical and acid depolymerization underwent dialysis at 4 °C for 48 h (cut-off of 500–1000 Da, Biotech CE Tubing, Spectra/Por^®^, Fisher Scientific, Illkirch, France).

These two depolymerization procedures led to the production of oligosaccharide fractions named DEP-HD PP-UE and DEP-AD PP-UE for radical hydrolysis by H_2_O_2_ and acid hydrolysis by ion-exchange resin procedure, respectively.

All produced fractions (crude, dialyzed, and depolymerized) were freeze-dried (Alpha 1–4 LSC, Martin Christ Gefriertrocknungsanlagen GmbH, Osterode am Harz, Germany) and stored at 4 °C until use.

The molecular weight distribution was analyzed by high-performance size-exclusion chromatography (HPSEC, UHPLC Ultimate 3000, Thermo Fisher Scientific, Waltham, MA USA). The biochemical composition was assessed by colorimetric methods. The monosaccharide composition was determined by high-performance anion-exchange chromatography with pulsed amperometric detection (HPAEC-PAD, Dionex^TM^ ICS-5000^+^ DC, Thermo Fisher Scientific, Illkirch, France), and completed with matrix-assisted laser desorption/ionization time-of-flight mass spectrometry (MALDI-TOF, Microflex ToF machine, Bruker Daltonik GmbH, Leipzig, Germany) [13].

### 4.3. Bacterial Strains and Culture Conditions

#### 4.3.1. *Staphylococcus aureus* and *Staphylococcus epidermidis* Strains

*Staphylococcus aureus* MFP03 and *Staphylococcus epidermidis* MFP04 were initially isolated from the skin of healthy volunteers [53]. The bacteria, stored at −140 °C in cryobead form, were initially plated on tryptic soy agar (TSA, Sigma-Aldrich, Saint-Quentin Fallavier, France). After overnight (O/N) incubation in static aerobic conditions at 37 °C, the Petri plate could be stored at 4 °C before use within one month. Colonies were transferred in a 10 mL tryptic soy broth (TSB) medium into a 100 mL sterile Erlenmeyer flask (1:10 ratio for aerobic conditions). These pre-cultures were incubated O/N under agitation at 180 rpm (rotations per minute) and 37 °C.

#### 4.3.2. *Cutibacterium acnes* Strains

The acneic strain RT4 HL045PA1/HM-516 and the non-acneic strain RT6 HL110PA3/HM-554 of *Cutibacterium acnes*, initially isolated by Fitz-Gibbon et al. (2013) [37], were diffused by BEI Resources American Type Culture Collection (Virginia, USA). These strains refer to phylotypes IA_1_ and II, respectively [35]. The bacteria, stored at −140°C in cryobead form, were initially plated on agar brain heart infusion (BHI) for RT6 and on agar reinforced clostridial medium (RCM, Sigma-Aldrich, Saint-Quentin Fallavier, France) pastone +2% glucose for RT4. As these strains grew under anaerobic conditions, the plates were incubated under anoxic static conditions at 37 °C for 72 h, using BD GasPack^TM^ EZ pouch systems (BD).

Colonies were transferred into sterile conical 15 mL plastic tubes (Falcon) filled to maximal capacity (15 mL) with RCM polypeptone +2% glucose for RT6 and RCM pastone +2% glucose for RT4. These pre-cultures were incubated in anoxic static conditions at 37 °C for 72 h.

### 4.4. Bacterial Growth Kinetic

The effects of poly- and oligosaccharide fractions on bacterial growth were evaluated by considering generation time, which corresponds to the necessary time for the bacteria to double in number. The generation time (G) of bacterial strains was calculated with Equation (1) and expressed in minutes for *S. aureus* MFP03 and *S. epidermidis* MFP04 and in hours for *C. acnes* RT4 and RT6:

(1)$$G\;(\mathrm{time}\;\mathrm{unit})=\frac{\ln2}{\mathrm{Slope}\;\mathrm{of}\;\mathrm{the}\;\mathrm{growth}\;\mathrm{curve}\;\mathrm{in}\;\mathrm{exponential}\;\mathrm{phase}\;}\times 100.$$

#### 4.4.1. *Staphylococcus aureus* and *Staphylococcus epidermidis* Strains

After pre-culture, *S. aureus* MFP03 and *S. epidermidis* MFP04 were collected (stationary phase) and cultured at an initial OD_580nm_ (optical density at 580 nm) = 0.08 in TSB, supplemented, or not (control condition), with poly- and oligosaccharide *Ulva* sp. fractions at 1000 μg/mL, in a Bioscreen HoneyComb plate (Thermo Fisher Scientific, Waltham, MA, USA). OD measurement was performed automatically in a microplate reader (Bioscreen C, Oy Growth Curves Ab Ltd., Helsinki, Finland) at 580 nm every 15 min for 24 h with continuous shaking.

#### 4.4.2. *Cutibacterium acnes* Strains

After pre-culture, *C. acnes* strains were collected (stationary phase) and cultured at an initial OD_580nm_ = 0.08 in RCM pastone +2% glucose (RT4) or RCM polypeptone +2% glucose (RT6) supplemented, or not (control condition), with poly- and oligosaccharide *Ulva* sp. fractions at 1000 μg/mL, in a 96-well flat-bottomed polystyrene plate (NUNC) under anoxic static conditions. For C. acnes RT4, the microplate was sealed with silicone and parafilm before incubation, and OD was monitored in the microplate reader (TECAN Spark 10M, Männedorf, Switzerland) automatically every 2 h at 580 nm for 72 h. In the case of *C. acnes* RT6, the plate was incubated in a Whitley A85 Workstation for 72 h and OD measurement was performed manually every 2 h in the microplate reader (TECAN Spark 10M, Männedorf, Switzerland) at 580 nm.

### 4.5. Biofilm Formation Activity: Confocal Laser Scanning Microscopy

The bacteria were pre-cultured as previously described (Section 4.3.1 for *S. aureus* MFP03 and *S. epidermidis* MFP04, and Section 4.3.2 for *C. acnes* RT4 and RT6). Then, the bacteria at an initial OD_580nm_ = 0.08 in a specific medium (TSB for *S. aureus* MFP03 and *S. epidermidis* MFP04, RCM pastone +2% glucose for *C. acnes* RT4, and RCM polypeptone +2% glucose for *C. acnes* RT6), supplemented, or not (control condition), with poly- and oligosaccharide *Ulva* sp. fractions at 1000 μg/mL, were grown under aerobic conditions for 16 h for *S. aureus* MFP03 and *S. epidermidis* MFP04 (O/N under agitation at 180 rpm), and under anoxic static conditions for 72 h for *C. acnes* RT4 and RT6.

After incubation, the bacteria in stationary phase were collected by centrifugation (8000× *g* for 5 min at room temperature). The supernatant was removed, and the bacterial pellet was rinsed with sterile physiological water NaCl 0.9% (PS) in order to remove all traces of the culture medium and poly- or oligosaccharide fractions. The biomass was again centrifuged (8000× *g* for 5 min at room temperature) before being resuspended in PS (bacterial suspension) at OD_580nm_ = 0.1 for *S. aureus* MFP03 and *S. epidermidis* MFP04, and OD_580nm_ = 1 for *C. acnes* RT4 and RT6.

Next, aliquots of 200 μL of bacterial suspension of *S. aureus* MFP03 or *S. epidermidis* MFP04 and 1 mL of bacterial suspension of *C. acnes* RT4 or RT6, were transferred to the wells of 24-well-plates with flat glass bottoms (SensoPlate, Greiner Bio-One, Frickenhausen, Germany). Each bacterial suspension was tested in triplicate for each condition under investigation, using at least four independent experiments for each strain.

After distribution of bacterial aliquots, the plates were incubated at 37 °C for 2 h to allow primary adhesion, under static and aerobic conditions for *S. aureus* MFP03 and *S. epidermidis* MFP04, and under static and anoxic conditions for *C. acnes* RT4 and RT6 (Whitley A85 Workstation, Don Whitley Scientific, Bingley, United Kingdom). Next, the non-adhered bacteria were carefully removed, and a fresh culture medium was added: 200 μL/well of TSB +2% glucose for *S. aureus* MFP03 and *S. epidermidis* MFP04, 1 mL/well of RCM pastone +2% glucose for *C. acnes* RT4, and 1 mL/well of RCM polypeptone +2% glucose for C. acnes RT6. The plates were incubated at 37 °C for 46 h under static and aerobic conditions for *S. aureus* MFP03 and *S. epidermidis* MFP04, and at 37 °C for 70 h under static and anoxic conditions for *C. acnes* RT4 and RT6 (Whitley A85 Workstation, Don Whitley Scientific, Bingley, UK).

At the end of the incubation time, the culture medium was carefully removed from each well, and the bacteria were stained with SYTO9 Green Fluorescent Nucleic Acid Stain (Thermo Fisher Scientific, Waltham, MA, USA) diluted at ratio 1:5000 in PS (300 μL/well). The plates were incubated in the dark at room temperature for 15 min. The SYTO9 solution was removed from each well, and PS was added (300 μL/well), followed by the addition of 1 drop/well of ProLong^TM^ Diamond Antifade Mountant (Fisher Scientific, Illkirch, France). After 24 h of incubation at room temperature in the dark, the plates were stored at 4 °C for prior analysis within one month.

The biofilms were observed using an LSM 710 inverted confocal laser scanning microscope (Carl Zeiss^®^ Microscopy, Oberkochen, Germany) with an oil immersion objective (×63). Three-dimensional (3D) images and orthocuts were obtained using the ZEN^®^ 2009 software package. The images are representative of the biofilm structure observed in a mean of four different fields in each well over the triplicate wells and four independent studies. The ImageJ software package with COMSTAT 2.1 plugin was used for image analysis [75], and the average biofilm thickness (μm) and biovolume (μm^3^/μm^2^) were determined.

### 4.6. HaCaT Keratinocytes Cell Culture and Infection

The HaCaT keratinocytes cell line (CLS, Cell Lines Services, 300493, Eppelheim, Germany) was obtained from the skin of a 62-year-old Caucasian male.

The keratinocytes were cultured in Dulbecco’s modified eagle medium (DMEM-F12, Lonza, Basel, Switzerland) supplemented with 10% (*v*/*v*) fetal bovine serum (FBS) and 1% antibiotic solution (*v*/*v*) of penicillin (100 IU/mL)/streptomycin (100 μg/mL), in a 25 cm^2^ flask, in a temperature-controlled incubator with 5% CO_2_ at 37 °C. The cells were sub-cultured by trypsinization (0.05%) and EDTA (ethylenediaminetetraacetic acid) solution after reaching 80% confluence. The cells were used between passages 30 and 45.

The cells were seeded onto 24-well plates at a final density of 2.5 × 10^5^ cells per well and grown until they reached 80% confluence. Twenty-four hours before infection with the bacteria, the cell medium was replaced by DMEM-F12 alone (an antibiotic and serum-free cell culture medium).

The bacteria grown in the presence or absence (control) of poly- and oligosaccharide *Ulva* sp.-derived EAE fractions at 1000 μg/mL were collected in the stationary phase and centrifuged (8000× *g* at room temperature for 5 min). The supernatants were removed, and the bacterial pellets were resuspended in PS. Another similar centrifugation was performed in order to remove any trace of the culture bacteria medium. The bacterial pellets were resuspended in PS, and OD_580nm_ was measured in order to infect HaCaT cells at an MOI (Multiplicity Of Infection, bacteria-to-cell ratio) of 10:1 for *S. epidermidis* MFP04 and *S. aureus* MFP03 for 16 h, and of 50:1 for *C. acnes* RT4 and RT6 for 18 h.

### 4.7. LDH Cytotoxicity Studies

The lactate dehydrogenase (LDH) assay is based on the measurement of LDH, a stable cytosolic enzyme, which is released upon cell lysis. LDH was measured in culture supernatants using the conversion of tetrazolium salt into a red formazan product. The amount of red formazan measured by spectrophotometry is proportional to the number of lysed cells.

The amount of LDH released by HaCaT keratinocytes was determined after 16 h of infection with *S. epidermidis* MFP04 or *S. aureus* MFP03 using the Pierce^TM^ LDH Cytotoxicity kit (88954, Thermo Fisher Scientific, Waltham, MA, USA). The LDH assay was performed following the supplier’s instructions. LDH maximum release control was performed by adding 10 μg/well of lysis buffer (Triton X100 0.1% in PS) 45 min before adding the reagent. At the end of the infection period (16 h, cells exposed to bacteria), 50 μL of the cell culture supernatant were transferred onto a new 96-well microplate (non-sterile). Next, 50 μL of reaction mixture were added per well, and solutions were mixed by gentle tapping. The microplate was incubated at room temperature in the dark for 30 min. The reaction was stopped by adding 50 μL/well of stop solution, and the solutions were mixed by gentle tapping. LDH release was measured with a microplate reader (TECAN Spark 10M, Männedorf, Switzerland) by subtracting the 680 nm absorbance value (background signal from instrument) from the absorbance value at 490 nm. Experiments were carried out on three biological replicates.

### 4.8. Interleukin-8 Secretion Study

The inflammatory response of HaCaT keratinocytes to acneic (RT4) and non-acneic (RT6) *C. acnes* strains grown in RCM (pastone and polypeptone, respectively) +2% glucose supplemented, or not supplemented, with poly- or oligosaccharide *Ulva* sp. fractions at 1000 μg/mL, was evaluated by assaying interleukin-8 (IL-8) secretion into the cell culture medium.

The amount of IL-8 released by HaCaT keratinocytes was determined using the Human IL-8 ELISA kit (KHC0081, Thermo Fisher Scientific, Waltham, MA, USA). After 18 h of infection (i.e., cells exposed to bacteria), the cell culture supernatants were collected and stored at −80 °C prior to analysis. The samples to analyze were diluted at 1:10 ratio in a sample buffer supplied by the manufacturer beforehand. The dosage was performed following the manufacturer’s instructions. Thus, 50 μL of sample, control, or Human IL-8 Standard (0 to 1000 pg/mL) were added per well. Then, 50 μL of Human IL-8 Biotin Conjugate were added into each well. After mixing by gentle tapping, the plate was incubated at room temperature for 1 h and 30 min. Next, the solution was thoroughly aspirated, and the wells were washed four times with 1X wash buffer. Next, 100 μL of 1X streptavidin-HRP solution were added into each well, and the plate was incubated at room temperature for 30 min. Again, the solution was thoroughly aspirated, and the wells were washed four times with 1X wash buffer. Next, 100 μL of stabilized chromogen were added into each well, and the plate was incubated at room temperature in the dark for 30 min. Finally, 100 μL of stop solution were added into each well, the plate was taped on the side to mix, and the absorbance was read with a microplate reader at 450 nm (Varioskan Lux, Thermo Fisher Scientific, Vantaa, Finland).

### 4.9. Statistical Analysis

All values are expressed as means of *n* experiments ± standard error (SE). Statistical analyses were performed using Addinsoft (2020). Shapiro–Wilk’s test was used to evaluate the normal distribution of the data. Significant differences between control and experimental samples were analyzed by Student’s *t*-test (independent, two-sided) when a normal distribution occurred and by Mann–Whitney’s test (non-parametric test) when the data were not normally distributed. Statistically significant differences compared with control are indicated by asterisks and related to the alpha value: * *p* < 0.05 and ** *p* < 0.01. Differences compared with control that are not statistically significant are characterized by the absence of asterisks on the figures and named “NS” in the main text.

## 5. Conclusions

This study demonstrates for the first time that the growth of representative skin bacteria *S. epidermidis*, *S. aureus*, and *C. acnes* is not modified in the presence of poly- and oligosaccharide *Ulva* sp.-derived EAE fractions at 1000 μg/mL. In addition, the fractions did not induce a modification of the cytotoxic potential of staphylococci towards keratinocytes. The authors suggest that the modulation of the biofilm formation activity of *S. epidermidis* MFP04 and *C. acnes* RT4 and RT6 by oligosaccharide *Ulva* sp. fractions could provide an advantage in cutaneous dysbiosis conditions such as acne. The effects of these oligosaccharide fractions should be investigated at the initial adhesion step of the biofilm formation process. The study also reveals that poly- and oligosaccharide fractions have an anti-inflammatory potential towards *C. acnes* strains infecting keratinocytes. Hence, these biological activities could be used to promote the restoration of skin homeostasis in the case of inflammatory processes in skin pathologies. The authors suggest that these sulfated rhamnose-rich seaweed molecules could be used for dermo-cosmetic or dermo-therapeutic applications. Further investigations are required, such as ex vivo and in vivo studies using clinical tests, to complete the demonstration and document the safety of the potential uses.

## Figures and Tables

**Figure 1 molecules-26-04763-f001:**
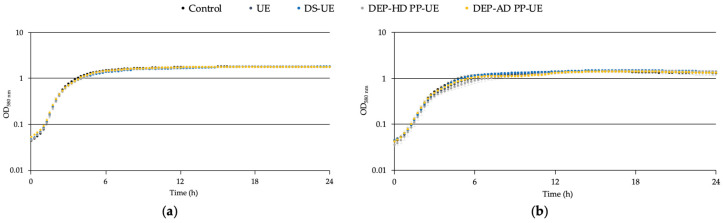
Effects of poly- and oligosaccharide *Ulva* sp. fractions at 1000 μg/mL on the growth kinetic of (**a**) *S. aureus* MFP03 (*n* = 3) and (**b**) *S. epidermidis* MFP04 (*n* = 3). Representation of the growth kinetic curve and error standard of each measurement are in log_10_ scale.

**Figure 2 molecules-26-04763-f002:**
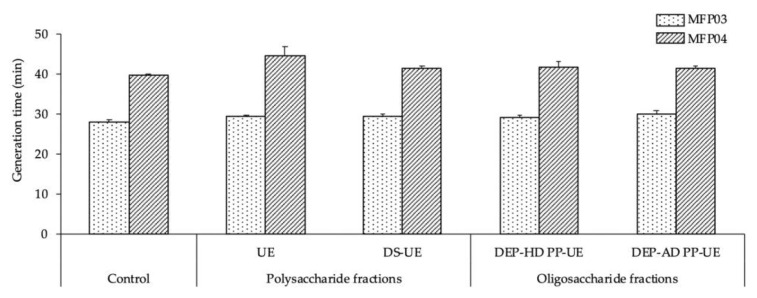
Effects of poly- and oligosaccharide *Ulva* sp. fractions at 1000 μg/mL on the generation time of *S. aureus* MFP03 (*n* = 3) and *S. epidermidis* MFP04 (*n* = 3). No statistically significant differences were observed in comparison with the control.

**Figure 3 molecules-26-04763-f003:**
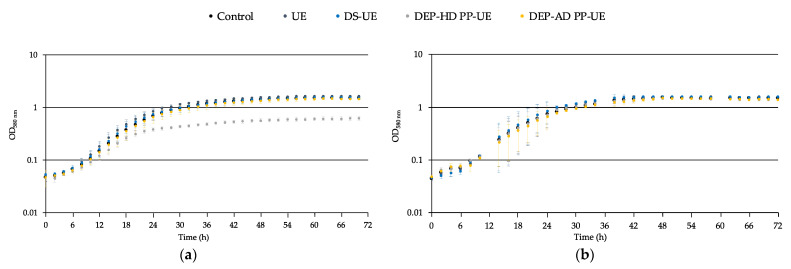
Effects of poly- and oligosaccharide *Ulva* sp. fractions on the growth kinetic of (**a**) *C. acnes* RT4 and (**b**) *C. acnes* RT6. The representation of the growth kinetic curve and error standard of each measurement are in log_10_ scale.

**Figure 4 molecules-26-04763-f004:**
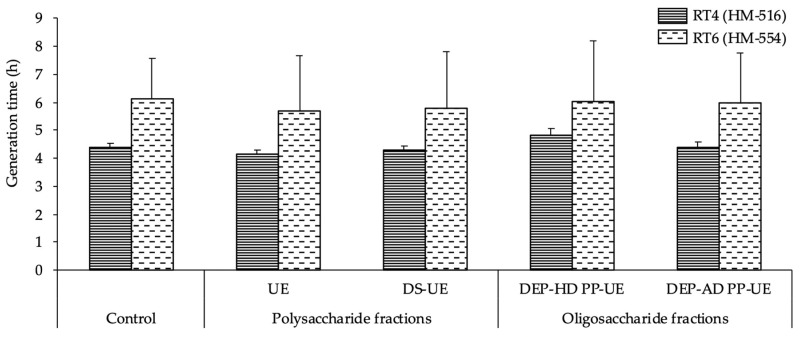
Effects of poly- and oligosaccharide *Ulva* sp. fractions at 1000 μg/mL on the generation time of *C. acnes* RT4 and *C. acnes* RT6. No differences were observed between control and treated bacteria.

**Figure 5 molecules-26-04763-f005:**
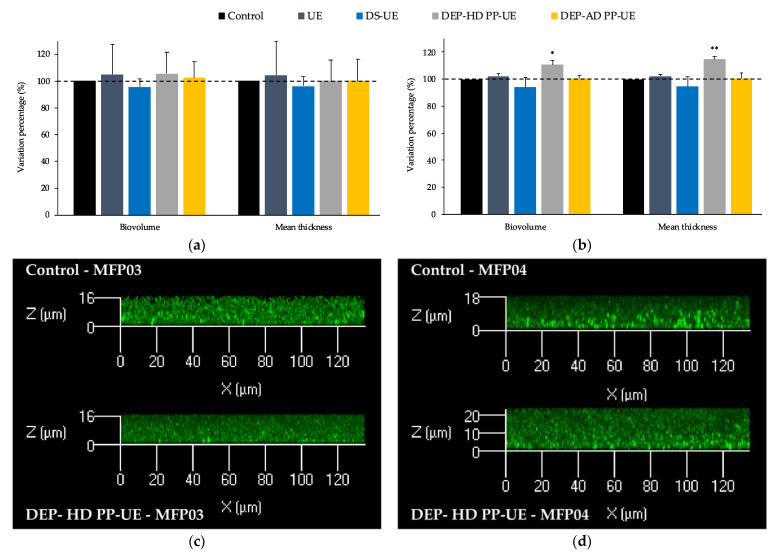
Effects of poly- and oligosaccharide *Ulva* sp. fractions at 1000 μg/mL on the biofilm production of (**a**) *S. aureus* MFP03 (*n* = 4) and (**b**) *S. epidermidis* MFP04 (*n* = 4) evaluated by confocal laser scanning microscopy. Significant differences between control (bacteria preliminarily cultured in TSB alone) and treated bacteria preliminarily cultured in TSB in the presence of the fractions are indicated by * *p* < 0.05 and ** *p* < 0.01. (**c**) Representative views of X/Z sections of the biofilms formed in control conditions (MFP03 preliminarily cultured in TSB alone) and when MFP03 were preliminarily cultured in TSB in the presence of DEP-HD PP-UE at 1000 μg/mL. (**d**) Representative views of X/Z sections of the biofilms formed in control conditions (MFP04 preliminarily cultured in TSB alone) and when MFP04 were preliminarily cultured in TSB in the presence of DEP-HD PP-UE at 1000 μg/mL.

**Figure 6 molecules-26-04763-f006:**
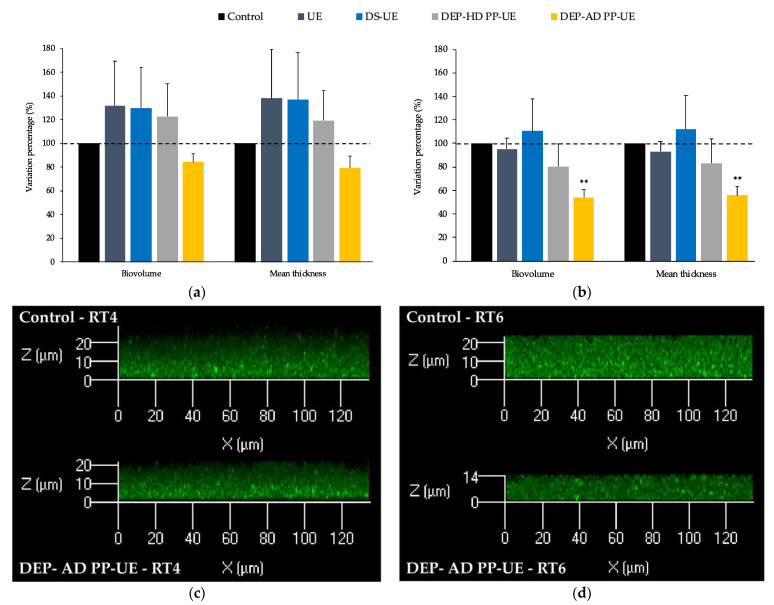
Effects of poly- and oligosaccharide *Ulva* sp. fractions at 1000 μg/mL on the biofilm production of (**a**) *C. acnes* RT4 (*n* = 4) and (**b**) *C. acnes* RT6 (*n*= 4) evaluated by confocal laser scanning microscopy. Significant differences between the control (bacteria preliminarily cultured in medium alone) and after the bacteria were preliminarily cultured in a cultured medium in presence of the fractions are indicated by ** *p* < 0.01. (**c**) Representative views of X/Z sections of the biofilms formed in control conditions (RT4 preliminarily cultured in RCM pastone +2% glucose alone) and when RT4 were preliminarily cultured in RCM pastone +2% glucose in the presence of DEP-AD PP-UE at 1000 μg/mL. (**d**) Representative views of X/Z sections of the biofilms formed in control conditions (RT6 preliminarily cultured in RCM polypeptone +2% glucose alone) and when RT6 were preliminarily cultured in RCM polypeptone +2% glucose in the presence of DEP-AD PP-UE at 1000 μg/mL.

**Figure 7 molecules-26-04763-f007:**
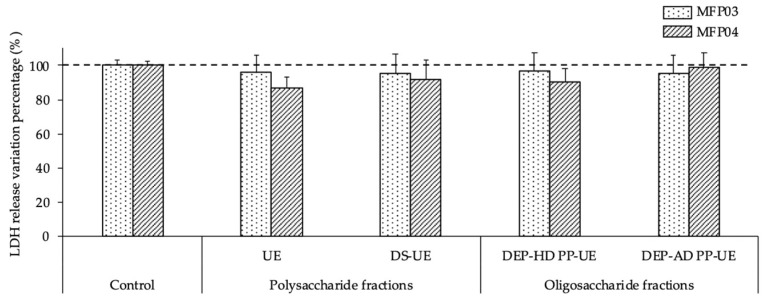
Effects of poly- and oligosaccharide *Ulva* sp. fractions at 1000 μg/mL on the cytotoxic potential of *S. aureus* MFP03 (*n* = 3) and *S. epidermidis* MFP04 (*n* = 3) towards HaCaT keratinocytes evaluated by LDH (lactate dehydrogenase assay). No statistically significant differences were observed in comparison to the control.

**Figure 8 molecules-26-04763-f008:**
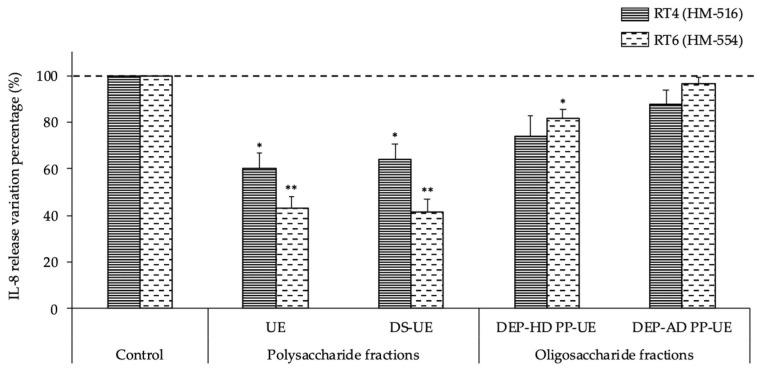
Effects of poly- and oligosaccharide *Ulva* sp. fractions at 1000 μg/mL on the inflammatory potential of *C. acnes* RT4 (*n* = 3) and *C. acnes* RT6 (*n* = 3) towards HaCaT keratinocytes evaluated by IL-8 ELISA assay. Significant differences between the control (bacteria preliminarily cultured in medium alone) and after the bacteria were preliminarily cultured with fractions are indicated by * *p* < 0.05 and ** *p* < 0.01.

**Figure 9 molecules-26-04763-f009:**
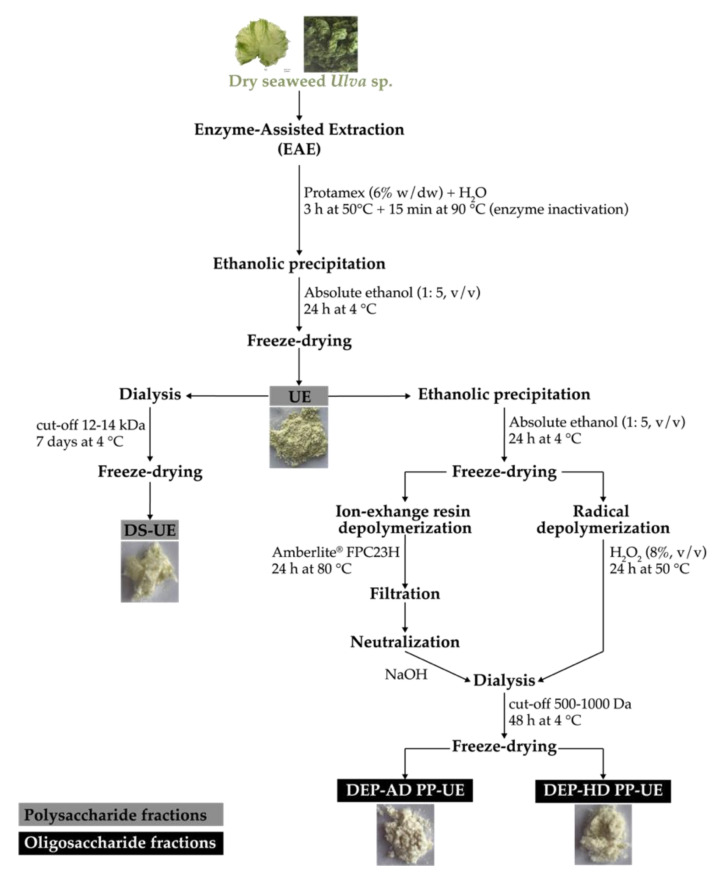
Detailed procedures for poly- and oligosaccharide *Ulva* sp.-derived EAE fractions [16].

**Table 1 molecules-26-04763-t001:** Description of poly- and oligosaccharide *Ulva* sp.-derived EAE fractions and their obtention process [16].

	Fraction Name	Description
Polysaccharide fractions	UE	Fraction rich in crude ulvans
	DS-UE	Dialyzed fraction from crude ulvans
Oligosaccharide fractions	DEP-HD PP-UE	Depolymerized fraction rich in ulvans obtained by radical H_2_O_2_ hydrolysis
	DEP-AD PP-UE	Depolymerized fraction rich in ulvans obtained by acid resin hydrolysis

## Data Availability

The data presented in this study are available on request from the corresponding author without restriction.

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
