# Peer review of "Effects of Ulva sp. Extracts on the Growth, Biofilm Production, and Virulence of Skin Bacteria Microbiota: Staphylococcus aureus, Staphylococcus epidermidis, and Cutibacterium acnes Strains"

_molecules, 2021, doi:10.3390/molecules26164763_

Round 1

Reviewer 1 Report

The manuscript presents how poly- and oligosaccharide fractions extracted from Ulva sp. affect the growth, biofilm production, and virulence of three strains of skin bacteria. The manuscript is very well written and presents thoughtful analyses. I believe that this paper will bring the great impact into marine metabolites studies. Therefore, I recommend it for the publication in Molecules, after minor revision (please unify the reference list and check if the strains names are written in italics in the whole manuscript).

Reviewer 2 Report

This study deals with the growth of "skin" bacteria, S. epidermidis, S. aureus, and C. acnes, in the presence of poly- and oligosaccharide Ulva sp. The authors are using state-of-art concepts to prove their hypotheses. The team (at least some of them) is publishing about Ulva sp. in the last years, addressing other aspects of the action of the Ulva sp. (those poly and oligosachrides) on the skin.

In my point of view, the main text is well structured, the results are relevant and the discussion is well done. Hence, it deserves to be published as presented.

Reviewer 3 Report

The submitted manuscript was highly duplicated from

Fournière, M., Bedoux, G., Lebonvallet, N., Leschiera, R., Goff-Pain, L., Bourgougnon, N., & Latire, T. (2021). Poly-and Oligosaccharide Ulva sp. Fractions from Enzyme-Assisted Extraction Modulate the Metabolism of Extracellular Matrix in Human Skin Fibroblasts: Potential in Anti-Aging Dermo-Cosmetic Applications. Marine Drugs19(3), 156.

Round 2

Reviewer 3 Report

 Accepted in present form